# Effects of Image Processing Using Honeycomb-Removal and Image-Sharpening Algorithms on Visibility of 27-Gauge Endoscopic Vitrectomy

**DOI:** 10.3390/jcm11195666

**Published:** 2022-09-26

**Authors:** Kuniharu Tasaki, Tomohisa Nishimura, Taro Hida, Kazushi Maruo, Tetsuro Oshika

**Affiliations:** 1Department of Ophthalmology, Faculty of Medicine, University of Tsukuba, 1-1-1 Tennoudai, Tsukuba 305-8575, Japan; 2Mikawa Eye Clinic, 4-3-1 Matsubara, Saga 840-0831, Japan; 3Department of Biostatistics, Faculty of Medicine, University of Tsukuba, 1-1-1 Tennoudai, Tsukuba 305-8575, Japan

**Keywords:** endoscopic vitrectomy, image-processing, visibility, 27-gauge

## Abstract

Endoscopic vitrectomy with small gauge probes has clinical potentials, but intraocular visibility is inherently limited by low resolution and dim illumination due to the reduced number of optic fibers. We investigated whether honeycomb-removal and image-sharpening algorithms, which enable real-time processing of live images with a delay of 0.004 s, can improve the visibility of 27-gauge endoscopic vitrectomy. A total of 33 images during endoscopic vitrectomy were prepared, consisting of 11 original images, 11 images after the honeycomb-removal process, and 11 images after both honeycomb-removal and image-sharpening procedures. They were randomly presented to 18 vitreous surgeons, who rated each image on a 10-point scale. The honeycomb-removal algorithm almost completely suppressed honeycomb artifacts without degrading the background image quality. The implementation of image-sharpening algorithms further improved endoscopic visibility by optimizing contrast and augmenting image clarity. The visibility score was significantly improved from 4.27 ± 1.78 for the original images to 4.72 ± 2.00 for the images after the honeycomb-removal process (*p* < 0.001, linear mixed effects model), and to 5.40 ± 2.10 for the images after both the honeycomb-removal and image-sharpening procedures (*p* < 0.001). When the visibility scores were analyzed separately for 10 surgeons who were familiar with endoscopic vitrectomy and 8 surgeons who were not, similar results were obtained. Image processing with honeycomb-removal and image-sharpening algorithms significantly improved the visibility of 27-gauge endoscopic vitrectomy.

## 1. Introduction

Optimal visualization of intraocular structures is one of the most important and sometimes challenging aspects of vitreoretinal surgery. When compromised anterior segment transparency or inadequate pupillary dilatation restricts transpupillary observation of the posterior eye segment, an adjunctive skill set of endoscopic vitrectomies can be of help [1,2]. In addition, the endoscope can visualize the ciliary processes, pars plana, and anterior retina without scleral depression, facilitating surgery on these areas without causing peripheral lens aberration or alteration of the normal anatomy from scleral indentation/deformation that are commonly encountered during conventional vitrectomy. The endoscope has been adopted in the management of various posterior eye segment pathologies, such as retinal detachment, ischemic retinopathies with neovascular glaucoma, severe ocular trauma, endophthalmitis, lens-related disorders in the posterior segment, and pediatric vitreoretinal diseases [3,4].

Due to the caliber constraints of intraocular probes, however, endoscopy comes with several limitations, including low resolution, narrow field of view, and dim illumination. This is especially true with smaller-gauge systems; the cross-section area of a 27-gauge probe is only 20% of that of a 20-gauge probe. Furthermore, images obtained with fiberscope-type endoscopes contain black fine mesh noise (dark honeycomb-like artifacts), which originates from the opaque cladding layer surrounding each single fiber in the image conductor [5,6,7]. Such artifacts adversely affect visibility during endoscopic procedures.

Recently, new technologies have been developed and implemented to improve the live endoscopic images by honeycomb-removal and image-sharpening algorithms with a delay of 0.004 s [8]. Clinical and experimental studies demonstrated that application of such image-processing procedures could improve dacryoendoscopic visibility and thus potentially increase the diagnostic accuracy and treatment outcomes in patients with lacrimal passage obstruction [8]. We conducted the current study to assess whether honeycomb-removal and image-sharpening algorithms can increase the visibility in small-gauge endoscopic vitrectomy.

## 2. Materials and Methods

### 2.1. Endoscopic Images

Endoscopic vitrectomy was performed by a single surgeon (K.N.) using a 27-gauge endoscope probe of 0.4 mm in diameter having 5000 pixels (VIT-27MFY-S; MACHIDA Endoscope Co., Ltd., Chiba, Japan) and a high-definition camera (MVH-2010A; MACHIDA Endoscope Co., Ltd., Chiba, Japan) at Mikawa Eye Clinic in February 2022.

For the purpose of comparative evaluation, still images were captured from the endoscopic video footage recorded during surgery. Eleven surgical scenes were selected for the evaluation, including observation of retinal and vitreous pathologies, laser photocoagulation, retinal view under air infusion, and inspection of anterior retinal region such as ora serrata. The freeze-frame images were first processed with the honeycomb-removal algorithm, followed by the image-sharpening algorithm. Thus, a total of 33 images were subjected to visual evaluation, consisting of 11 original images, 11 images after the honeycomb-removal process, and 11 images after both the honeycomb-removal and image-sharpening procedures.

The Institutional Review Board of Tsukuba University Hospital approved the study protocol (R03-101). This study was conducted in accordance with the Declaration of Helsinki. Each patient gave an informed consent in a written form before undergoing surgery.

### 2.2. Image Processing

The endoscopic images were processed using the honeycomb-removal (WipeFiber^®^; Logic & Design Inc., Tokyo, Japan, and MACHIDA Endoscope Co., Ltd., Chiba, Japan) and the image-sharpening (Medical Image Enhancer: MIEr^®^; Logic & Design Inc., Tokyo, Japan) algorithms. The devices containing these proprietary algorithms were connected to the endoscopy systems, and real-time image processing of video footages with a delay of 0.004 s was performed. In the current study, however, the image-processing procedures were implemented on the freeze-framed pictures captured from the video in order to assess the effects of algorithms on identical source images.

The honeycomb-removal technology was originally developed for improving the imaging quality of endoscopes for other organs, such as gastrointestinal endoscopes, which was then applied to dacryoendoscopes [8,9]. The noise caused by the honeycomb structure appears as a high-frequency component in the spatial frequency domain and can be removed by a low-pass filter. In this study, we used a Gaussian filter, which is a high-performance and an easy-to-implement low-pass filter. Subsequently, the image was refined by applying contour enhancement only in the specific frequency region according to the fiberlet diameter, while simultaneously suppressing the restoration of the removed honeycomb structure.

Following the honeycomb-removal procedure, the image-sharpening process, consisting of two steps, was used to further improve the visibility of endoscopic images. First, image contrast was enhanced by optimizing the local dynamic range depending on the contrast level of neighbouring small areas; the dynamic range can be expanded in high- and low-contrast areas independently. Second, the degraded resolution of images was restored by mathematically decomposing frequencies and narrowing the point spread function, without employing artificial interpolation or manipulation of image data.

### 2.3. Image Visibility Evaluation

The endoscopic images were evaluated by 18 vitreous surgeons, including 10 surgeons who were familiar with endoscopic vitrectomy and 8 surgeons who were not. The evaluators were presented with 33 images in a random order, one at a time. The images were presented on a 13.3-inch monitor (resolution of 2560 × 1600 pixels), and the images, which were more than 15 cm in diameter, were sequentially displayed on a black background. Each image was scored on a 10-point scale for visibility.

### 2.4. Statistical Analysis

Numerical data are presented as a mean ± standard deviation unless otherwise noted. The results were statistically analyzed and compared among three groups using the linear mixed effects model. Statistical significance was set at *p* < 0.05. All statistical analyses were performed using SAS software (version 9.4, SAS Institute, Cary, NC, USA) and SPSS (version 27; IBM, Armonk, NY, USA).

## 3. Results

Figure 1 shows the original images, images after honeycomb-removal process, and images after both honeycomb-removal and image-sharpening procedures of representative cases. In the original images, honeycomb-structure artifacts were inevitably observed, which hinder the visibility of the intraocular structures (Figure 1, upper row). The honeycomb-removal algorithm almost completely eliminated honeycomb artifacts without degrading the image quality (Figure 1, middle row). Implementation of image-sharpening algorithms further improved visibility by optimizing contrast and augmenting the image clarity (Figure 1, bottom row).

The results of image visibility evaluation are shown in Figure 2. The visibility score was significantly improved from 4.27 ± 1.78 for the original images to 4.72 ± 2.00 for the images after the honeycomb-removal process (*p* < 0.001), and to 5.40 ± 2.10 for the images after both honeycomb-removal and image-sharpening procedures (*p* < 0.001). When the visibility scores were analyzed separately for 10 surgeons who were familiar with endoscopic vitrectomy and eight surgeons who were not, similar results were obtained (Figure 3).

## 4. Discussion

The most visually conspicuous, and disturbing artifact, arising from the image transmission through coherent fiber bundles, is the honeycomb effect. The honeycomb effect is a consequence of the light being guided from the distal to the proximal end of the individual cores comprising the fiber bundle but not through the surrounding cladding. Each core, while usually imaged across multiple pixels, contains intensity information on a single, discrete position within the imaged scene. Consequently, the resulting raw image data is a high-resolution rectangular matrix representation of a low-resolution, irregularly sampled scene [9]. In this study, we used a Gaussian filter, which is a high-performance and easy-to-implement low-pass filter, for honeycomb-removal. As shown in Figure 1, all original images (A–E) exhibited evident honeycomb artifacts which hindered clear observation of intraocular structures. By implementing the honeycomb-removal algorithm, the honeycomb artifact was almost completely suppressed (Figure 1F–J), resulting in significantly higher visibility scores rated by 18 evaluators. Band-pass filtering provides a simple and efficient approach to suppressing/removing the honeycomb structure from fibered endoscopic images. However, given the irregularly distributed cores in most modern miniaturized fiberscopes, identifying suitable thresholds in the frequency domain that would remove the honeycomb effect (usually a high frequency component) without blurring the underlying imaged structure (usually a lower frequency component) can be inherently challenging [9]. The fact that our approach successfully removed the honeycomb artifact without degrading the image quality indicates that selection and application of the Gaussian filter in our study were appropriate.

The 27-gauge endoscopic vitrectomy represents the smallest-caliber probe system currently available on the market. The 27-gauge endoscopes can fit through standard small-gauge vitreoretinal surgery microcannulas, thereby increasing their utility in modern vitreoretinal surgery. Smaller diameter endoscope probes, however, are produced by reducing the number of optical fibers, leading to lower optical resolution and dimmer illumination which make surgical manipulation in complex vitreoretinal diseases such as severe proliferative vitreoretinopathy difficult. Thus, the introduction of a post-procedural solution to mathematically improve endoscopic visibility would be more beneficial for smaller-caliber endoscopy systems.

In the present study, proprietary algorithms were applied to enhance image quality by optimizing contrast and restore resolution by narrowing the point-spread-function. It was shown that the addition of these image-processing procedures to the honeycomb-removal process further improved endoscopic visibility. The post-processing images depicted small retinal pathologies such as holes and degeneration more clearly, made images under air infusion brighter, and enabled more detailed observation of structures at the extreme periphery. We found that surgeons who were not familiar with endoscopic vitrectomy rated the quality of original images lower than those familiar with endoscopic vitrectomy. This was probably because the former surgeons judged that the 27-gauge endoscopic images had poorer quality in comparison with the images under the conventional surgical microscopes. Despite this, a similar degree of improvement in the visibility score was reported by both groups of surgeons. This result may indicate that their judgements on the effects of image processing were unbiased and robust regardless of their previous experience of endoscopic vitrectomy.

The merits of the current technologies include that real-time processing is possible with a delay of only 0.004 s, where surgeons do not perceive any lag time during intraoperative maneuvers. In addition, the algorithms can be easily integrated into an existing endoscopic system. The device containing the algorithms is simply connected between the camera and monitor, and real-time image processing starts immediately.

There are several limitations to this study. First, the current study was not designed to evaluate surgical outcomes directly, but the visibility of endoscopic images was analyzed. Although we expect that improvement in image quality would lead to better treatment results, such assumptions should be tested in future large-scale clinical studies. Second, other related factors such as the length of treatment and the degree of surgeon stress associated with intraoperative visibility, which could be additional potential benefits of the current technologies, were not assessed in this study.

## 5. Conclusions

The visibility of 27-gauge endoscopic vitrectomy was significantly improved by image processing, including the elimination of honeycomb artifacts, optimization of contrast, and restoration of resolution.

## Figures and Tables

**Figure 1 jcm-11-05666-f001:**
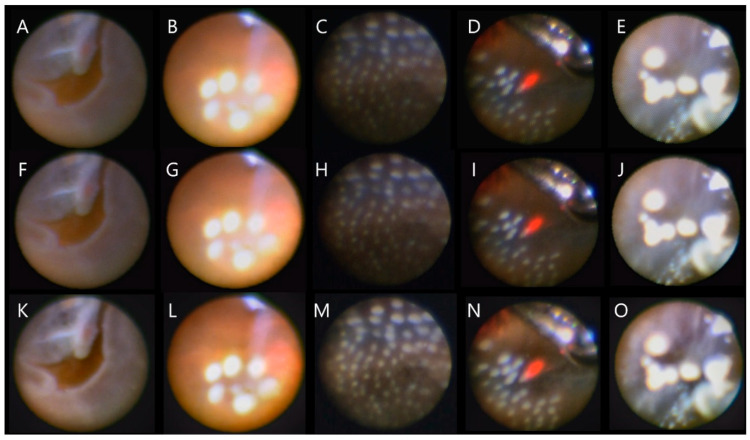
Original images (**A**–**E**), images after honeycomb-removal process (**F**–**J**), and images after both honeycomb-removal and image-sharpening procedures (**K**–**O**). Image processing significantly improved endoscopic visibility, including more detailed observation of lattice degeneration next to a retinal tear (**K**), a small retinal hole surrounded by laser photocoagulation (**L**), a retinal break and laser responses under air infusion (**M**), retinal lesions at ora serrata (**N**), and asteroid hyalosis (**O**).

**Figure 2 jcm-11-05666-f002:**
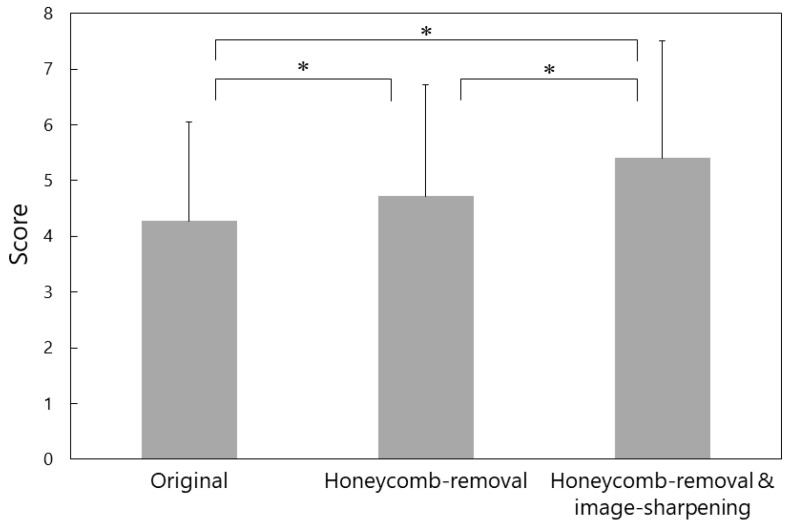
Visibility scores for 33 images of endoscopic vitrectomy assessed by 18 evaluators. The score was significantly improved from 4.27 ± 1.78 for the original images to 4.72 ± 2.00 for the images after honeycomb-removal processing (* *p* < 0.001, linear mixed effects model), and to 5.40 ± 2.10 for the images after both honeycomb-removal and image-sharpening procedures (* *p* < 0.001).

**Figure 3 jcm-11-05666-f003:**
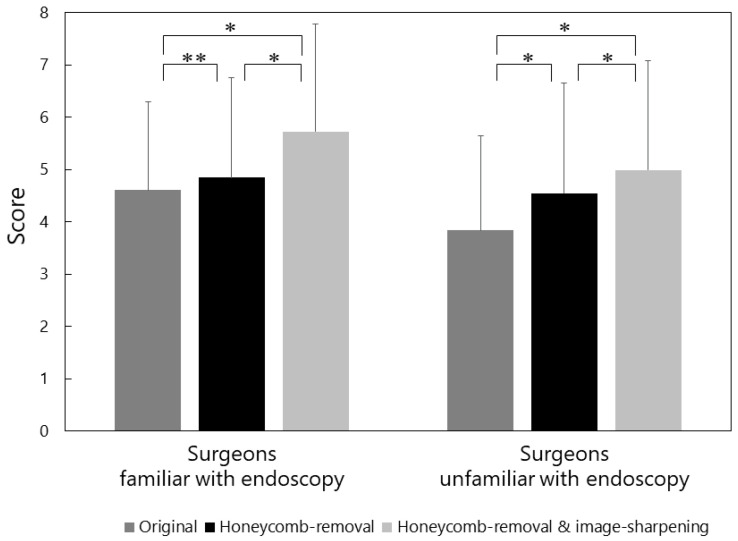
Visibility scores analysed separately for surgeons who were familiar with endoscopic vitrectomy (*n* = 10) and those who were not (*n* = 8). Implementation of image-processing algorithms significantly improved visibility of endoscopic vitrectomy (* *p* < 0.001, ** *p* = 0.076, linear mixed effects model).

## Data Availability

The datasets generated during and/or analyzed during the current study are available from the corresponding author on reasonable request.

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
