# Peer review of "Effects of Image Processing Using Honeycomb-Removal and Image-Sharpening Algorithms on Visibility of 27-Gauge Endoscopic Vitrectomy"

_jcm, 2022, doi:10.3390/jcm11195666_

Round 1

Reviewer 1 Report

Very good method to improve the image with the 27 endoscope. 

Author Response

Thank you for the positive comment.

Reviewer 2 Report

Kuniharu Tasaki et al, submitted “Effects of image processing using comb-removal and image-2 sharpening algorithms on visibility of 27-gauge endoscopic 3 vitrectomy” as an original article.

Congratulations on your impressive work and images.

-       -  Can you mention if the 18 surveyed participants were retina surgeons? If not, what is there subspeciality.

-        - Line 169 “currently available on the market”, I think it is available only in some countries, I may be mistaken. If only available in Japan, can you mention that.

-        - In line 200, can clarify what is meant by degree of surgeon stress? Do you mean experience?

 Thank you for the opportunity to review this article, I appreciate it.

Author Response

Congratulations on your impressive work and images.

Can you mention if the 18 surveyed participants were retina surgeons? If not, what is there subspeciality.

Response: As mentioned in the original manuscript, there were 18 vitreous surgeons, including 10 surgeons who were familiar with endoscopic vitrectomy and 8 surgeons who were not (line 108).

Line 169 “currently available on the market”, I think it is available only in some countries, I may be mistaken. If only available in Japan, can you mention that.

Response: Thank you for the comment. It is true that 27-gauge endoscopic vitrectomy is not available in all countries, but the statement “27-gauge endoscopic vitrectomy represents the smallest caliber probe system currently available on the market is correct.

In line 200, can clarify what is meant by degree of surgeon stress? Do you mean experience?

Response: The phrase “the surgeon stress” has been replaced by “the degree of surgeon stress associated with intraoperative visibility” (line 204).